# TensorJSFuzz: Effective Testing of Web-Based Deep Learning Frameworks via Input-Constraint Extraction

## Abstract

As web applications grow in popularity, developers are increasingly integrating deep learning (DL) models into these environments. Web-based DL frameworks (e.g., TensorFlow.js) are essential for building and deploying such applications. Ensuring the quality of these frameworks is critical for the reliability of DL systems. While extensive testing efforts have been made for native DL frameworks such as TensorFlow and PyTorch, web-based DL frameworks have not yet undergone systematic testing. A key challenge in this context is generating high-quality inputs that are both syntactically and semantically valid, as well as designing effective test oracles tailored to the unique constraints of web-specific environments. To address this gap, we introduce TensorJSFuzz, a novel method for testing web-based DL frameworks. To ensure input quality, TensorJSFuzz extracts constraints directly from the source code of framework APIs. By leveraging Large Language Models (e.g., ChatGPT) to understand the code and extract input constraints, TensorJSFuzz performs type-aware random generation coupled with dependency-aware refinement to create high-quality test inputs. These inputs are then subjected to differential testing across various backends, including CPU, TensorFlow, Wasm, and WebGL. Our experimental results show that TensorJSFuzz outperforms baseline methods in generating valid inputs and identifying bugs. In particular, TensorJSFuzz successfully detected 92 bugs, with 30 already confirmed or fixed by developers, demonstrating its effectiveness in improving the robustness of web-based DL frameworks.

## Keywords

Web-based Deep Learning, Fuzzing, Large Language Model

**ACM Reference Format:**
Anonymous Author(s). 2018. TensorJSFuzz: Effective Testing of Web-Based Deep Learning Frameworks via Input-Constraint Extraction. In *Proceedings of Make sure to enter the correct conference title from your rights confirmation emai (Conference acronym 'XX).* ACM, New York, NY, USA, 9 pages. https://doi.org/XXXXXXX.XXXXXXX

## 1 Introduction

Deep learning (DL) has gained widespread application in diverse fields, including image classification [23, 25], natural language processing [19, 35], and speech recognition [16, 20]. Traditionally, DL models have been deployed using native deep learning frameworks

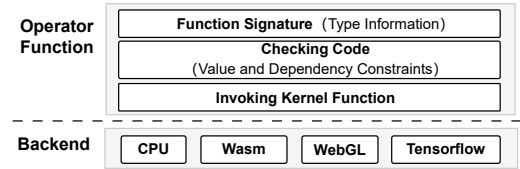

**Figure 1: The code structure of DL operator in Tensoflow.js**

like TensorFlow and PyTorch, which are optimized for desktop and server environments. However, with web applications increasingly simplifying cross-platform portability issues and gaining popularity, developers are integrating DL models into web applications more often. Web-based DL frameworks (e.g., TensorFlow.js) are crucial for the development and deployment of such applications, offering a wide array of functional operators, and allowing developers to deploy DL models directly within web browsers.

The quality and reliability of these web-based DL frameworks are paramount, as they directly impact the overall performance and dependability of web-based DL models and applications. Unlike their native counterparts, web-based frameworks are constrained by the inherent limitations of the browser environment, such as restricted access to memory and hardware accelerators. To mitigate these constraints, web-based DL frameworks employ a range of acceleration mechanisms, including WebAssembly and WebGL, which introduce new challenges for testing DL frameworks in the web environment. Compared to the testing of native DL frameworks, testing web-based frameworks must account for the variability of web environments. These include browser implementations, hardware variability, and the intricacies of web technologies like WebAssembly, which presents both a performance benefit and a source of potential bugs.

A key challenge in testing web-based frameworks is generating high-quality test cases that thoroughly explore the logic of core APIs. Specifically, DL operators (or APIs) often require inputs in the form of high-dimensional tensors with complex interdependencies. As a result, randomly generated inputs frequently fail the operator's validation checks, limiting their ability to effectively test core functionality. To address this, FreeFuzz [34] mines test cases from open-source repositories. DocTer [36] uses rule-based approaches to collect constraints from API function descriptions in the documentation. ACETest [31] specifically collects constraints from C++ code. However, these approaches often struggle to generate effective test cases due to unclear constraints, missing or inaccurate API descriptions, or being tailored for native DL frameworks.

To address these challenges, we propose TensorJSFuzz, the first fuzzer specifically designed for web-based DL frameworks, such as TensorFlow.js [1]. As shown in Figure 1, a typical web-based operator consists of three key components: the *function signature*, input

---

[1]We focus on TensorFlow.js in this paper as it is currently the most popular web-based DL framework, and other web-based DL frameworks do not yet offer accessible APIs. However, our approach can be generalized to other web-based DL frameworks.

validation (*checking code*), and a backend-specific *kernel function*. Our goal is to generate inputs that bypass the validation checks and thoroughly test the kernel function. To achieve this, TensorJSFuzz infers the parameter types and the constraints on them, which are critical for generating valid and effective test inputs.

Specifically, TensorJSFuzz begins by analyzing the Abstract Syntax Tree (AST) [28] of the function signature to extract parameter type information. Next, to identify dependency constraints between parameters in the validation checks, TensorJSFuzz leverages the capabilities of Large Language Models (LLMs) [13], utilizing their understanding of code through in-context learning to extract these constraints. Based on the inferred types and constraints, we design a heuristic-based approach for input generation, which includes type-aware random generation and dependency-aware input refinement. To account for the multiple backend implementations used by web-based frameworks, TensorJSFuzz also incorporates differential testing across various backends (as shown in Figure 1), making that inputs not only bypass validation checks but also trigger potential inconsistencies between different backends.

We evaluated TensorJSFuzz on TensorFlow.js, where it successfully extracted 2,046 constraints from 187 selected operators. These constraints included 1,426 type constraints and 620 dependency constraints. To assess the effectiveness of TensorJSFuzz, we compared it against three representative baselines: a random input generator (Random), a native DL fuzzer (DocTer), and an SMT-based approach (TensorJSFuzz-SMT). The experimental results show that TensorJSFuzz significantly outperforms the baselines in generating valid inputs and identifying bugs. Specifically, TensorJSFuzz generated 71.83% valid inputs, compared to 36.05% for Random, 38.79% for DocTer, and 62.12% for TensorJSFuzz-SMT. Additionally, TensorJSFuzz identified 64 unique bugs that neither Random nor DocTer were able to detect. In total, TensorJSFuzz uncovered 92 bugs, with 30 of them already confirmed or fixed.

In summary, this paper makes the following contributions:

- We present TensorJSFuzz, the first testing framework specifically designed for the TensorFlow.js library, representing a significant advancement in ensuring the reliability and robustness of web-based DL frameworks.
- We introduce a novel approach that leverages LLMs to extract input constraints directly from the source code of DL operators. Additionally, we propose a generation technique that includes *type-aware input generation* and *dependency-aware input refinement*, enabling the effective generation of diverse and high-quality test inputs.
- We demonstrate the effectiveness of TensorJSFuzz through comprehensive comparative experiments with existing DL fuzzers. TensorJSFuzz successfully uncovered **92** bugs, with 30 already confirmed or fixed.
- The source code and experimental data are publicly available at [8] for further research and replication.

## 2 Background and Motivation

### 2.1 Preliminary

TensorFlow.js [18] is a leading web-based DL framework, enabling seamless integration of DL models into web applications. It provides a versatile platform for developing and deploying models directly

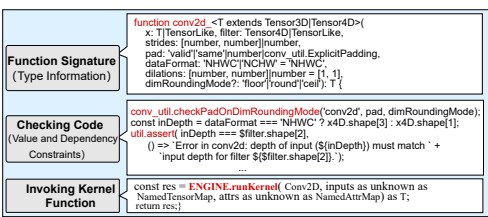

Figure 2: The source code of tf.conv2d

in web browsers. TensorFlow.js supports model training and inference on diverse backends, providing flexibility and performance optimizations for different environments. The library comprises various backends, including CPU [4], WebGL [7], Wasm [6], and the TensorFlow [5]. Each backend caters to different hardware and execution contexts, contributing to TensorFlow.js's adaptability and widespread use in web-based deep learning applications.

### 2.2 Motivation Example

The key insight of our approach that extracts constraints from source code is from the structured code in Web-based DL frameworks. As illustrated in Figure 2, the source code of the `tf.conv2d` operator comprises three key components: the function signature, checking code, and the invocation of the kernel function.

The function signature explicitly defines the types for each parameter. For instance, the parameter *x* is designated as Tensor3D or Tensor4D, indicating a tensor of rank 3 or 4. The checking code employs assertions or functions to check the syntactical and semantical validity of parameters. A notable example from the checking code in Figure 2 is the dependency between the parameters *dataFormat*, *x*, and *filter*. If *dataFormat* is NHWC, then *x.shape[3]* must match *filter.shape[2]*. Otherwise, *x.shape[1]* should equal *filter.shape[2]*.

## 3 Approach

Figure 3 presents an overview of TensorJSFuzz, which contains three stages. The initial stage involves constraint extraction, where TensorJSFuzz extracts two types of constraints from a DL operator's source code: (1) type information for each parameter, derived from the function signature's abstract syntax tree, and (2) dependency constraints, extracted from the function body using LLMs. The type information includes the structure, data type, rank, and enumerated values of each parameter, while dependency constraints cover the permissible range of parameter values and their interdependencies.

Based on the constraints, TensorJSFuzz aims to generate valid inputs. Initially, TensorJSFuzz randomly generates inputs that align with the extracted type information, ensuring type consistency. These inputs are then refined and adjusted to meet the dependency constraints, significantly enhancing the likelihood of input validity.

TensorJSFuzz further employs three test oracles to identify various bug types, including crash, memory-related, and logic bugs. Specifically, logic bugs are detected by differential testing across different backends. For memory-related bug detection, particularly in the Wasm backend, TensorJSFuzz utilizes AddressSanitizer.

### 3.1 Constraint Extraction

*3.1.1 Type Information Extraction.* The function signature provides detailed syntax information for each input parameter, such as the data structure, data type, and enumerated values, which can

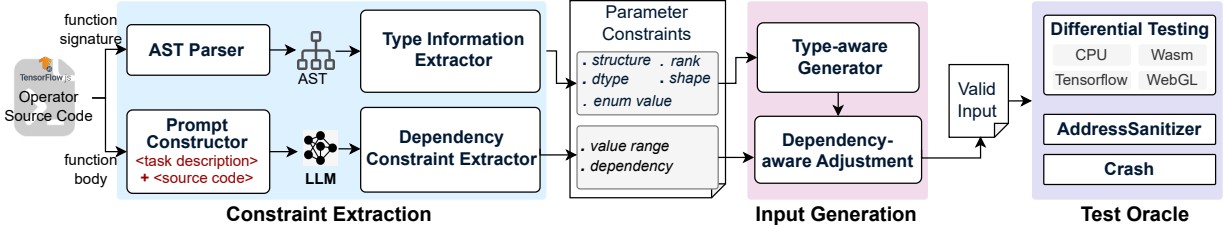

**Figure 3: Overview of TensorJSFuzz**

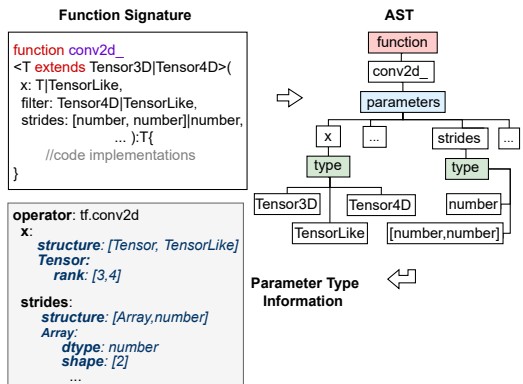

**Figure 4: The example of extracting type information**

be used to constrain the input generation. Therefore, we design a type information extractor to extract such type information from the function signature. Specifically, for each DL operator, the type information extractor first parses its function signature into an abstract syntax tree (AST). This AST is a tree with multiple typed nodes, where the root node represents the operator function and the 'parameters' node encapsulates details about all parameters of the operator. Each child node of the node 'parameters' represents a parameter. Within each parameter node, there is a 'type' node storing all the syntax details. The type information extractor subsequently retrieves syntax information from the 'type' node for each parameter and refines it into our type information representation. To facilitate the subsequent input generation phase, we categorize the type-related constraints into the following five types:

- *structure*: the data structure that stores a collection of values for the input parameter, such as tuple, array, and tensor.
- *rank*: the number of dimensions of a tensor/array.
- *shape*: the shape of the tensor/array.
- *dtype*: the data type, such as number, boolean, int, and string, of the parameter or the element type of the tensor/array.
- *enum value*: a set of valid values.

Figure 4 shows an example of extracting type information for the parameters of `tf.conv2d` operator. The type information extractor parses it into an abstract syntax tree (i.e., AST in Figure 4), where the 'parameters' node and 'type' node are marked as the blue box and green box, respectively. Following this, the extractor acquires syntax information from the 'type' node for each parameter and further refines it into type information based on categories. For example, the obtained syntax information of parameter *strides* is "[number, number]|number", and the refined type information are *{structure:[Array, number], dtype: number, shape: [2]}*.

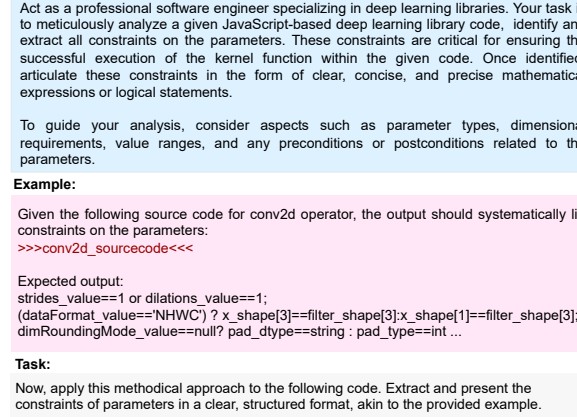

**Figure 5: The prompt for querying ChatGPT**

*3.1.2 Dependency Constraint Extraction.* To ensure input validity, knowing only the type information is insufficient, as the constraints, such as value ranges and inter-parameter dependencies, can have high influence in the input validity. For instance, parameters often have specific valid value ranges. Moreover, their data type, rank, or values may depend on other parameters. Such detailed constraints are discernible only through an in-depth analysis of the source code (i.e., the checking code). To capture this information, we introduce a specialized extractor for extracting information about the value range and parameter dependencies from the checking code.

Considering the complexity of code like tensor calculations and diverse conditional checks, we leverage LLMs, known for their exceptional comprehension in both natural language processing and code-related tasks [10, 11, 14, 27, 38]. In this work, ChatGPT [1] was chosen for constraint extraction using a one-shot prompting strategy. Figure 5 shows an example of this approach, where the prompt includes a task description and specific example. This example illustrates the expected output relative to the task.

Table 1 presents a selection of constraint examples extracted by ChatGPT, covering four distinct types. The second row, for example, highlights a *rank* constraint, specifying that the *rank* of the *indices* parameter must be greater than or equal to the *batchDims* parameter value in the `tf.gather` operator. The third row illustrates a *shape* constraint, where the fourth dimension of *x* must match the third dimension of *filter*. Additionally, *dtype* and *value* constraints are shown in the fourth and fifth rows, respectively, indicating dependencies of one parameter's dtype or value on another.

**Table 1: Examples of constraints extracted by ChatGPT**

| Type | Operator | Constraint |
|------|----------|-----------|
| **rank** | tf.gather | indices_rank>=batchDims_value |
| **shape** | tf.conv3d | x_shape[4]==filter_shape[3] |
| **dtype** | tf.add | a_dtype==b_dtype |
| **value** | tf.conv3d | strides_value==1 or dilations_value==1 |

---

**Algorithm 1:** Type-aware Input Generation

---

**Input** : $\mathcal{T}$: Type information of all parameters of operator
**Output**: $RI$: Randomly generated inputs

1   $\mathcal{P} := \text{getParameters}(\mathcal{T})$;
2   **for** $p \in \mathcal{P}$ **do**
3     $structure := \text{randomSelect}(\mathcal{T} \to p \to structure)$;
4     **if** $\text{isAtomicType}(structure)$ **then**
5       **if** $\text{hasEnumValue}(\mathcal{T} \to p)$ **then**
6        $RI \to p := \text{randomSelect}(\mathcal{T} \to p \to enum\_value)$;
7       **else**
8        $dtype := \text{randomSelect}(\mathcal{T} \to p \to dtype)$;
9        $RI \to p := \text{randomGenerate}(dtype)$;
10    **else**
11      $rank := \text{randomSelect}(\mathcal{T} \to p \to rank)$;
12      $shape := \text{randomSelect}(\mathcal{T} \to p \to shape)$;
13      $dtype := \text{randomSelect}(\mathcal{T} \to p \to dtype)$;
14      $RI \to p := \text{generate}(rank, shape, dtype)$;
15   **return** $RI$;

---

## 3.2 Input Generation

To generate diverse inputs that conform to the constraints. A direct approach would involve using a Satisfiability Modulo Theories (SMT) [15] solver to compute inputs on the extracted constraints. However, existing works [26, 31, 33] have highlighted limitations of SMT solvers in generating diverse inputs, as they typically produce boundary values and face challenges in solving constraints related to tensors, such as the high costs associated with solving constraints on a tensor's value. Therefore, we developed a lightweight and heuristic-based method to generate valid inputs, which unfolds in two primary steps: (1) type-aware input generation, and (2) dependency-aware input adjustments.

*3.2.1 Type-aware Input Generation.* Leveraging the type information extracted from the function signature (see Figure 4), TensorJS-Fuzz initiates the input generation process. This involves randomly generating an input for each parameter while meticulously considering its type information. Algorithm 1 presents the details for random input generation. Given the extracted type information (i.e., $\mathcal{T}$) of all parameters, TensorJSFuzz first obtains the parameter list (i.e., $\mathcal{P}$) (Line 1). Next, it randomly selects the structure for each parameter from the structure list specified in the type information (Line 3). If the selected structure is atomic and the enumerated values are specified in the type information, the parameter value is randomly chosen from those values (Lines 5 to 6). Otherwise, it chooses a dtype and generates a random value based on the chosen dtype for the parameter with atomic structure (Lines 7 to 9). If the selected structure is not atomic, TensorJSFuzz further selects the rank, shape, and dtype for the parameter and randomly generates a value based on them (Lines 10 to 14). Finally, we obtain a random input that satisfies the type constraints (Line 15).

*3.2.2 Dependency-aware Input Adjustments.* To ensure that generated inputs satisfy dependency constraints, we introduce a dynamic

```
<constraint> ::= <expression>
<expression> ::= <term>
        | <expression> <operator> <expression>
        | '(' <expression> ')'
        | <expression> ?< expression >: <expression>
<term> ::= <value> | <variable>
<value> ::= <number> | <string> | <int> | <float>
<operator> ::= <arithmetic_operator> | <logical_operator> | <comparison_operator>
<arithmetic_operator> ::= '+' | '-' | '*' | '/' | '%'
<logical_operator> ::= 'or' | 'and' | 'not'
<comparison_operator> ::= '<' | '>' | '>=' | '<=' | '==' | '!='
```

**Figure 6: The constraint BNF grammar**

---

**Algorithm 2: Adjust**

---

**Input** : $C$: A set of constraints on all parameters
       $RI$: Randomly generated inputs
**Output**: $CI$: Adjusted inputs

1   $CI := RI$;
2   **for** $c \in C$ **do**
3    **if** $\text{isLogicalExpression}(c)$ **then**
4      **if** $c.op = \text{'or'}$ **then**
5       $LR := \textbf{Adjust}(\{c.left\}, CI)$;
6       **if** $LR = CI$ **then**
7        $\textbf{Adjust}(\{c.right\}, CI)$;
8      **else if** $c.op = \text{'and'}$ **then**
9       $\textbf{Adjust}(\{c.left\}, CI)$;
10      $\textbf{Adjust}(\{c.right\}, CI)$;
11    **else if** $\text{isCMPExpression}(c)$ **then**
12      **if** $\text{notSatisfy}(c, CI)$ **then**
13       $LR := \textbf{AdjustParam}(c.left, c, CI)$;
14       **if** $LR = CI$ **then**
15        $\textbf{AdjustParam}(c.right, c, CI)$;
16   **return** $CI$;
17   **Function** $\textbf{AdjustParam}(exp, c, CI)$
18    **if** $\text{isRank}(exp)$ **then**
19      $\text{updateValidRank}(exp, c, CI)$;
20    **if** $\text{isDtype}(exp)$ **then**
21      $\text{updateValidDtype}(exp, c, CI)$;
22    **if** $\text{isShape}(exp)$ **then**
23      $\text{updateValidShape}(exp, c, CI)$;
24    **if** $\text{isValue}(c)$ **then**
25      $\text{updateValidValue}(exp, c, CI)$;
26    **return** $CI$;

---

adjustment strategy that iteratively modifies inputs until all constraints are met. To achieve this, a parser capable of recognizing the extracted constraints is necessary. We manually reviewed the constraints gathered by ChatGPT and summarized them into a simplified constraint syntax, as depicted in Figure 6. In this context, the term *variable* refers to various parameter characteristics, including rank, shape, value, or data type.

Our adjustment algorithm shown in Algorithm 2, takes as input a set of constraints $C$ and random inputs $RI$, producing adjusted inputs $CI$ likely satisfying the constraints. The algorithm functions as a parser, interpreting the constraint syntax and applying necessary modifications for each constraint $CI$ (Lines 2 to 15). When encountering an *or* logical expression (Line 4), the algorithm attempts to adjust the left-hand side (Line 5) and, if unsuccessful (Line 6), the right-hand side (Line 7). For *and* logical expressions, both sides are adjusted (Lines 8 to 10). Note that expressions involving *NOT* or *ternary* logic can be transformed into equivalent expressions. For example, $\neg(a > b)$ can be converted to $a <= b$. The constraint $a == b?c.type == int : c.type == float$ can be converted to $(a == b \land c.type == int) \lor (a \neq b \land c.type == float)$.

For comparison expressions (Line 11) that do not satisfy constraints (Line 12), adjustments are made to the left-hand side (Line 13) or the right-hand side (Line 15), depending on the types of the parameters involved. Based on the comparison in *c*, for *rank* types (e.g., indices_rank==1), TensorJSFuzz tries to modify the rank (Line 19) of the parameter indices; for *dtype* or *shape* types (e.g., a_dtype==b_dtype), it tries to alter the data type or shape (Line 21 and Line 23); and for *value* types (e.g., stride_value==1), it directly changes the parameter value (Line 25), such that the constraints *c* can be satisfied. These modifications are based on the left or right operators of the comparison expressions. For instance, consider a random input *RI* for the operator `tf.conv2d`. Suppose the values of parameters *strides* and *dilations* are [3,5] and [4,7], respectively. They meet the type constraints but break the dependency constraint $strides\_value == 1 \ or \ dilations\_value == 1$. An adjustment is necessary to make them comply, typically modifying *strides* or *dilations* to [1,1].

It is important to note that, given the undecidability of the constraint-solving problem, the heuristic-based method in Algorithm 2 is not a perfect solver. Constraints that contain syntax errors generated by the LLMs, unsupported syntax elements, or adjustments that fail to resolve properly will result in the algorithm returning the original, unadjusted inputs (as seen in Line 16 and Line 26). Consequently, some inputs may not be successfully adjusted by Algorithm 2.

### 3.3 Test Oracle

To systematically capture bugs during testing, TensorJSFuzz incorporates the following four test oracles:

**Memory Bugs**: Utilizing AddressSanitizer [3], TensorJSFuzz detects memory-related bugs within Wasm backend, a context where memory safety is not guaranteed. AddressSanitizer is adept at identifying a spectrum of memory bugs, such as memory out-of-bounds, memory leaks, and use-after-free errors, bolstering our capability to uncover memory bugs.

**Crash Bugs**: We characterize crash bugs as any abrupt terminations of the program, including unexpected exceptions, aborts, and segmentation faults. Similar to previous work [34], we also employ heuristic methods to filter the expected exceptions which are typically syntax-related exceptions, caused by invalid inputs.

**Differential Testing**: For identifying logical bugs (Wrong Computation Bugs) that do not disrupt execution, we conduct differential testing across four TensorFlow.js backends: CPU, WebGL, Wasm, and TensorFlow. When the same input produces divergent outputs from operators across these backends, a bug is suspected. To account for minor discrepancies, which may arise from backend-specific computational precision and are not considered bugs, we apply the following metric:

$$difference = \frac{\sum_{i=1}^{N} |A_i - B_i|}{N}$$

where $N$ is the total number of output tensor elements, and $A_i$, $B_i$ represent the i-th elements of tensors A and B, respectively. A difference exceeding a predefined threshold indicates a potential wrong-computation bug. In this paper, to avoid false positives caused by the natural and expected differences between different backends, we set a larger threshold of 1,000.

## 4 Evaluation

To evaluate the effectiveness of TensorJSFuzz, we aim to answer the following research questions:

**RQ1:** How effective is TensorJSFuzz in accurately extracting constraints from the source code of web-based DL frameworks?

**RQ2:** How does TensorJSFuzz perform in generating inputs and detecting bugs when compared to baselines?

**RQ3:** What kinds of bugs can be detected by TensorJSFuzz?

### 4.1 Experimental Setup

**Baselines**. For a comparative analysis in our study, we selected DocTer [36], the method most closely aligned with ours, which extracts constraints from API function descriptions, as the baseline. We excluded ACETest because it is specifically designed for C++ code. To ensure a fair comparison, we extracted API descriptions for TensorFlow.js operators from the official documentation, used DocTer's replication package to generate inputs, and integrated our testing oracles into DocTer.

Furthermore, we implemented 2 additional representative baselines: 1) *Random*, a type-aware random fuzzer that recognizes parameter types but ignores dependency constraints. 2) TensorJSFuzz-SMT, a variant of TensorJSFuzz, which translates constraints into SMT formulas and leverages Z3 for generating random solutions. As Z3 does not have a built-in batch sampling function, to obtain diverse solutions, we continuously insert new constraints to block the newly obtained solution. After this step, TensorJSFuzz-SMT can get a batch of unique solutions for the constraints. For parameters that do not have constraints, it randomly generates the values.

**Environment.** In our experiments, the model GPT-4 is used. To manage the randomness of ChatGPT's responses, we conducted experiments with various parameter settings. Based on our experience, we selected the optimal parameter values: the parameters *top_p* and *temperature* are set to 0.1 and 0.5, respectively. We tested TensorFlow.js on the version 4.1.0, which defines 231 DL operators in tfjs-core, divided into nine categories. Each operator was tested through a headless Chrome browser, facilitated by Puppeteer [2]. Since the browser was opened and closed three times for each test input across three backends: CPU, Wasm, and WebGL, the average processing time was approximately 3 seconds per input. To effectively manage the time constraints, we followed the approach of [36] and limited each fuzzer to produce 1,000 test inputs per operator. To mitigate the impact of randomness, each experiment was repeated three times during testing, and the average values of these runs were used for comparative analysis.

All experiments are conducted on a high-performance workstation equipped with a 64-bit Ubuntu 20.04 LTS system, 32GB RAM, and two 18-core 2.3GHz Intel Xeon E5-2699 CPUs.

### 4.2 RQ1: Effectiveness of constraint extraction

*4.2.1 The number of constraints.* Table 2 displays the number of constraints extracted by DocTer and TensorJSFuzz. The constraints extracted by TensorJSFuzz are composed of two main types. The row *Type Info* shows constraints related to type information. Meanwhile, *Den. Constraints* represents the number of dependency constraints identified, quantified as the total count of individual extracted expressions. Columns 3-6 indicate the number of constraints related to

**Table 2: Number of extracted constraints**

| | Constraint Type | dtype | structure | shape | value | Total |
|---|---|---|---|---|---|---|
| **DocTer** | Type & Den. | 130 | 414 | 165 | 49 | **538** |
| **TensorJSFuzz** | Type Info | 423 | 500 | 327 | 176 | 1,426 |
| | Den. Constraints | 233 | 0 | 232 | 155 | **620** |
| | Total | 656 | 500 | 559 | 331 | **2,046** |

**Table 3: Quality of dependency constraints**

| | dtype | shape | value | Total |
|---|---|---|---|---|
| **Precision(%)** | 81.9 | 96.9 | 94.9 | 90.9 |
| **Recall(%)** | 94.5 | 94.2 | 97.7 | 95.2 |
| **F1(%)** | 87.8 | 95.5 | 96.1 | 93.3 |

each parameter. Given that rank equates to the length of the shape, rank-related constraints are grouped under the shape category.

TensorJSFuzz extracts a total of 2,046 constraints, nearly four times more than DocTer, which is 538. TensorJSFuzz is more effective than DocTer, especially in terms of the shape and value properties. Structure-related constraints can be expressed in simple natural language, so DocTer can also easily obtain such constraints from the documents, which leads to similar constraint numbers of structure in the table. In particular, TensorJSFuzz extracts 620 dependency constraints, whereas most of the constraints extracted by DocTer are limited to type constraints due to its lack of code-level analysis. Additionally, we did not observe any structural constraints, as TensorFlow.js does not perform structure validation in its checking code. These results demonstrate that TensorJSFuzz is capable of automatically extracting more comprehensive constraints, significantly reducing the need for manual effort.

*4.2.2 The quality of extracted constraints.* Type information comes from function signatures via static methods and is precise. Meanwhile, ChatGPT provides dependency constraints. To assess the quality of these dependency constraints, we randomly selected 20% (95 parameters) for manual verification. This verification was conducted independently by this paper's three authors and resulted in unanimous agreement. For each parameter, we annotated specific constraints based on the source code to establish a solid ground truth. The constraints extracted by ChatGPT were then compared against this benchmark. In the verification, we employed standard metrics including precision, recall, and the F1 score. Precision represents the percentage of correctly extracted constraints (those matching the ground truth) out of all extracted constraints. Recall is the percentage of correctly extracted constraints out of the total ground truth constraints. The F1 score is the harmonic mean of precision and recall.

Table 3 displays the precision, recall, and F1 score for each category of dependency constraint. Overall, ChatGPT achieves a high precision (90.9%), recall (95.2%), and F1 score (93.3%) across all three categories. ChatGPT is more effective in extracting shape/value-related dependency constraints with an F1 score over 95%. It is less effective in dtype-related dependency constraints. The reason is that ChatGPT sometimes misinterprets "Tensor" as data type. For instance, it might extract a constraint like "x_dtype==Tensor". This does not affect the generation of valid inputs, as for these kinds of dependency-free dtype constraints, TensorJSFuzz adheres to the extracted *Type Info*.

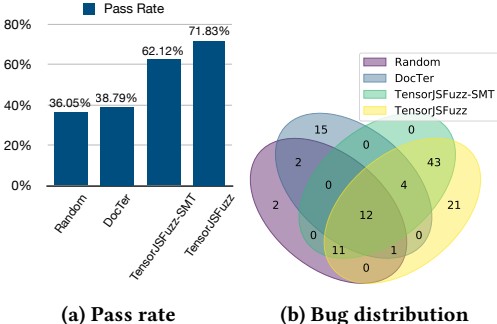

(a) Pass rate  (b) Bug distribution

**Figure 7: Comparison between TensorJSFuzz and baselines regarding pass rate and bug distribution**

> **Answer to RQ1:** Compared to DocTer, TensorJSFuzz is capable of extracting more constraints, and its precision and recall in constraint extraction are satisfactory.

## 4.3 RQ2: Comparison with existing approaches

*4.3.1 The effectiveness of generating inputs.* Generating valid inputs is essential for passing a DL operator's validity checks. Since manual input validation is impractical, we follow prior work [36] and consider inputs that terminate normally—i.e., without exceptions, as a reasonable approximation of validity. An input is deemed valid if it successfully terminates on any backend. We assess the ratio of passing inputs generated by each tool, i.e., pass rate. Note that DocTer can be configured to generate inputs that violate constraints. For a fair comparison, we set the *mutation_p* in DocTer to 0, ensuring only generates inputs that adhere to constraints.

Figure 7a shows the input pass rates for each tool. Notably, TensorJSFuzz achieves a 71.83% pass rate, surpassing Random (36.05%), DocTer (38.79%), and TensorJSFuzz-SMT (62.12%). This marks an increase of 199.25% over Random and 185.17% over DocTer, largely due to TensorJSFuzz's efficient extraction of dependency constraints. Moreover, despite TensorJSFuzz and TensorJSFuzz-SMT utilizing identical constraints, TensorJSFuzz records a higher pass rate. This discrepancy arises because TensorJSFuzz-SMT does not address constraints related to tensor values, the number of elements, or loop constraints due to their computational cost [31]. Properties corresponding to these unresolved constraints are generated randomly. These findings underscore the proficiency of TensorJSFuzz in generating valid inputs that effectively test core functionalities.

Further investigation into invalid inputs generated by TensorJSFuzz revealed some inaccuracies in constraints extracted by ChatGPT. For example, it fails to extract the implicit constraint $a\_shape == b\_shap$ in the operator *tf.add*, which are not checked in the JavaScript source code. Additionally, some invalid inputs stem from our syntax parsing's limitations. Specifically, certain complex scenarios, like array range indexing (e.g., $mask\_shape == tensor\_shape[axis : axis+mask\_rank]$) and data structure parameters (e.g., *HTMLVideoElement*), were not fully supported in TensorJSFuzz.

*4.3.2 The effectiveness of detecting bug.* We conducted a comparative analysis of TensorJSFuzz against all baselines for bug identification. Aligning with DocTer's optimal settings, which involve parameters like optional_ratio and mutation_p, we evaluated different configurations on a randomly selected 10% subset of operators

**Table 4: The number of bugs detected by different tools**

| Backend | CPU | Wasm | WebGL | Tensorflow | Total |
|---|---|---|---|---|---|
| **Random** | 6.00 | 7.67 | 7.67 | 3.33 | 24.67 |
| **DocTer** | 7.33 | 9.67 | 10.67 | 4.67 | 32.34 |
| **TensorJSFuzz-SMT** | 14.67 | 29.67 | 14.33 | 9.33 | 68.00 |
| **TensorJSFuzz** | 22.33 | 36.67 | 19.00 | 11.67 | 89.67 |

to find the best one. The configuration that yielded the best results in our tests set optional_ratio to 0.2 and mutation_p to 0.4.

Table 4 shows the number of bugs detected by each tool. The column 2-4 indicates the average total number of bugs found in each backend. We can see that TensorJSFuzz uncovered 89.67 bugs across the four backends. Notably, TensorJSFuzz outperformed each baseline, Random(24.67), DocTer(32.34), and TensorJSFuzz-SMT (68.00), in every backend.

On investigating the bugs that DocTer and Random failed to identify, we attributed this to their inability to extract complex dependencies. For instance, both Random and DocTer struggled to identify dependencies between parameters like *x* and *filter* in convolution operators, as outlined in Section 2. This led to only 1-2 out of 1000 inputs passing checks, greatly reducing test effectiveness. However, for the same constraints, TensorJSFuzz-SMT detects fewer bugs than TensorJSFuzz because the generated inputs are not diverse enough. We observed that TensorJSFuzz-SMT often generates boundary values, even when additional constraints are introduced after each iteration to encourage more diverse inputs. For instance, in the case of *tf.conv3d*, among the 1,000 generated inputs, tensor x had only *29* unique shapes. Additionally, variations in these shapes were limited to the first and last elements, resulting in shapes resembling "[,1,1,1]". Comparatively, TensorJSFuzz achieves higher diversities, for example, tensor x had *999* unique shapes in the case of *tf.conv3d*, which explore space of valid input more adequately. These findings underscore TensorJSFuzz's superior performance in bug detection, attributed primarily to its effective extraction of dependency constraints and valid input generation.

We also analyze the distribution of bugs found by each tool. As seen in Figure 7b, these tools find different bugs. Note that here we count the total number of bugs detected across all repetitions. For example, TensorJSFuzz can find all bugs found by TensorJSFuzz-SMT since they generate inputs using the same constraints. 64, 15, and 2 unique bugs are found by TensorJSFuzz, DocTer, and Random, respectively. This is due to the differences in their respective methods of extracting constraints. Random and DocTer miss 68 and 75 bugs found by TensorJSFuzz, respectively. This is because they cannot extract the fine-grained constraints. TensorJSFuzz misses 4 bugs found by Random due to the randomness of the input generation process. DocTer found some unique bugs because it can generate some inputs that violate constraints to test the checking code of DL operator. Differently, TensorJSFuzz mainly generates valid inputs conforming to constraints. However, TensorJSFuzz still detects more bugs than DocTer, highlighting the importance of generating valid inputs.

Additionally, we further compared the average time each tool takes to discover the first bug for each operator. Moreover, we recorded the input ID that triggered the first bug, indicating the number of inputs needed to trigger the first bug. The results are

**Table 5: Average time to find the first bug**

| | TensorJSFuzz | TensorJSFuzz-SMT | DocTer | Random |
|---|---|---|---|---|
| **#Inputs** | 290.75 | 415.75 | 687.65 | 809.32 |
| **Times(min)** | 14.54 | 34.64 | 34.38 | 41.20 |

**Table 6: Distribution of detected bugs by TensorJSFuzz**

| #Bugs (#Wrong-computation, #Crashes, #Memory) | | | | Total | Confirmed (Fixed) |
|---|---|---|---|---|---|
| CPU | Wasm | WebGL | Tensorflow | | |
| 23(8/15/0) | 37(10/2/25) | 20(8/12/0) | 12(4/8/0) | 92 | 30(11) |

presented in Table 5. We can observe that DocTer and Random take more than twice the time compared to TensorJSFuzz to discover the first bug. Moreover, on average TensorJSFuzz only needs to generate 290.75 inputs to discover a bug, while TensorJSFuzz-SMT, DocTer, and the Random require 415.75, 687.65, and 809.3 inputs, respectively. These results further indicate the TensorJSFuzz is more efficient in detecting bugs.

> **Answer to RQ2:** TensorJSFuzz generates more valid inputs than all baselines. Moreover, TensorJSFuzz demonstrates a notable advantage in both the efficiency and effectiveness of bug detection over all baselines.

## 4.4  RQ3: Bug Analysis

We further performed an in-depth analysis to characterize the bugs we detected. Table 6 presents detailed statistics about the bugs found by TensorJSFuzz. The number of wrong-computation bugs, crash bugs, and memory bugs are shown in "()" of the column #Bugs. We can observe that TensorJSFuzz detected 92 bugs in total (with 30 already confirmed as previously unknown bugs), and 11 of them have been fixed by the developers to date. The unconfirmed bugs are reproducible and waiting for the response of the developers.

The 92 bugs include 30 wrong-computation bugs, 37 crash bugs, and 25 memory bugs. Specifically, we can observe that most wrong-computation bugs (26/30) are distributed in the backend CPU, Wasm, and WebGL. 25 memory bugs are identified in the Wasm backend, respectively. No memory bugs are discovered in the CPU, WebGL and TensorFlow backends, which mainly arises from the absence of a dedicated memory bug oracle. These results indicate considerable inconsistencies in the implementation logic of TensorFlow.js operators across the four backends. In particular, the implementations for the web-specific backends, i.e., CPU, Wasm, and WebGL, should align with the mature Tensorflow backend, which invokes the same tensorflow.so as the DL framework TensorFlow.

In addition to detecting the three main categories of bugs mentioned above, we also uncovered 41 inconsistent behaviors between the Tensorflow backend and the other three web-specific backends. These discrepancies arise from variations in the supported parameter values. For example, when the parameter *pad* is set to a number, the operator of Tensorflow backend returns an exception with *"TF Backend supports only 'valid' and 'same' padding while padding was NUMBER"* while other backends return an output tensor. These inconsistencies, while not classified as bugs in our study, highlight shortcomings in the cross-platform deployment of TensorFlow.js.

**Case-Study 1 (Memory Bug):** Figure 8 shows the code that triggers a memory bug in the operator tf.conv2d. When running it in the Wasm backend, a memory error occurred with the message

```
var x=tf.ones([1,16,7,4]);
var filter =tf.fill([17,13,4,4],3,"float32");
var prediction = await tf.conv2d(input,filter,[25,24],-4,"NHWC",[1,1],"ceil");
```
**Target API: tf.conv2d**
**Catch:** requested allocation size exceeds maximum supported size.

**Figure 8: The example of memory bug**

```
var x=tf.fill([1,15,16,8],32,"float32");
var df=tf.fill([9,10,8,11],3,"float32");
var pf=tf.fill([1,1,88,6],3,"float32");
const result = tf.separableConv2d(input,df,pf,1,"valid", [0,2],"NHWC");
```
**wasm:**RuntimeError: null function or function signature mismatch
**Tensorflow:**Tensor[2280960,2280960,....]
**Target API: tf.separableConv2d**
**Catch:** Crash/Inconsistent between backends

**Figure 9: The example of crash bug**

```
var x = tf.fill([1,3,3,3,3],3,"float32")
    var result = tf.avgPool3d(x,[1,2,2], 1, 3,"floor","NDHWC");
// CPU result: [[[[NaN,NaN,NaN],[NaN,NaN,NaN],[NaN,NaN,NaN]....]]]
// Tensorflow result: [[[[0,0,0],[0,0,0],[0,0,0],....]]]]
```
**Target API: tf.avgPool3d**
**Catch:** Inconsistent between backends

**Figure 10: The example of wrong-computation bug**

*"requested allocation size 0xd55559f0 exceeds the maximum supported size of 0xc0000000"*. Debugging revealed that a negative *pad* was converted from *number* to *size_t* in the Wasm-specific kernel *wasmConv2d*, becoming *4294967292*, which caused *indirection_buffer_size* to exceed the allocation limit of *xnn_reallocate_memory*. The bug has been confirmed by developers. Since parameters *x*, *filter*, and *dataFormat* must meet the dependency constraint $DataFormat == NHWC?x\_shape[3] = filter\_shape[2] : x\_shape[1] = filter\_shape[2]$, Random and DocTer fail to detect this bug.

**Case-Study 2 (Crash Bug):** Figure 9 shows a crash bug in `tf.separableConv2d`. When running the code snippet on the backend Wasm, the crash is triggered with the message *"RuntimeError: null function or function signature mismatch"*. This crash bug has been confirmed by the developers who replied *"...I was able to replicate the issue. We'll investigate further and update soon..."*. Since parameters *x*, *depthwiseFilter*, and *pointwiseFilter* need to satisfy the dependency constraint $pointwiseFilter\_shape[2] === x\_shape[3] * depthwiseFilter\_shape[3]$, making Random and DocTer unable to detect the bug.

**Case-Study 3 (Wrong-Computation Bug):** Figure 10 shows a wrong-computation bug in `tf.avgPool3d`. When running the code snippet on the backend CPU, it returns a tensor with all elements set to NaN. However, the backend WebGL returns a tensor with all elements set to 0. The developers have fixed this bug by modifying the CPU-specific kernel function to avoid dividing zero when computing averages. All of the methods can detect this bug as it does not require complex dependency constraints.

---

**Answer to RQ3:** TensorJSFuzz detected 92 real-world bugs in total, 30 of which have been confirmed or fixed by developers.

---

## 5 Related Work

### 5.1 Model-level Fuzzing of DL Framework

Model-level fuzzers focus on generating various DL models for the target DL framework. CRADLE [29] is the first work to find and localize bugs in DL frameworks, which detects inconsistencies by running existing models on multiple backends of Keras.

LEMON [9] and AUDEE [22] further extend the idea of CRADLE to generate more diverse models. Muffin [21] generates DL models for testing DL frameworks in both the inference and training phases. Recently, NNSmith [26] tested DL compilers by generating diverse yet valid DNN models. These works all focus on fuzzing the native DL frameworks (e.g., TensorFlow and PyTorch). Different from them, we employ a more fine-grained operator-level fuzzing technique to test each operator of the web-based DL framework, i.e., Tensorflow.js.

### 5.2 Operator-level Fuzzing of of DL Framework

Operator-level fuzzing focuses on testing individual operators of the DL framework, which can test more operators than model-level fuzzing. FreeFuzz [34] mines inputs from open-source code snippets and then apply random mutations to generate diverse inputs. Similarly, SkipFuzz [24] employs an active learning approach, inferring the input constraints through the fuzzing process. DeepREL [12] and EAGLE [32] further leverage differential testing on relational operators (e.g., operators that always return the same results/statuses given the same inputs) to cover more operators. DocTer [36] extracts the input constraints from API documentation and then generates inputs based on these constraints. ACETEST [31] is specifically designed for native DL frameworks and extracted constraints from the code of the low-level DL operator specifically implemented with C/C++. More recently,∇Fuzz [37] utilizes automatic differentiation as the test oracle for more effective fuzzing. Different from the above model- and operator-level fuzzers, [17] apply modern Large Language Models (LLMs) [14] to generate diverse DL API sequences for testing.

While the aforementioned works are all effective in discovering bugs in DL frameworks, none of the existing fuzzing techniques targeted the web DL frameworks (e.g., Tensorflow.js). Different from them, firstly, we target the web-based DL framework, i.e., TensorFlow.js, which is different from native libraries in terms of the implementations of DL backends and the execution environments. Secondly, previous fuzzers extract input constraints from API documentation or infer valid input from open-source code snippets. We utilize the capabilities of Large Language Models (LLMs) to comprehend code and extract the dependency constraints via an in-context learning mechanism. Thirdly, We designed a new Oracle for the Wasm backend of Tensorflow.js, leveraging AddressSanitizer [30] to detect memory-related bugs, considering the characteristics of the web-based library.

## 6 Conclusion

This paper presents TensorJSFuzz, the first fuzzer specifically designed for testing web-based DL framework. TensorJSFuzz excels in extracting high-quality constraints, deriving type-related constraints from function signatures and dependency constraints directly from the function code. These constraints allow TensorJSFuzz to generate valid inputs that bypass syntactical checks, improving the effectiveness of testing within the web environment. Our evaluation demonstrates that TensorJSFuzz significantly outperforms existing baselines in detecting bugs both effectively and efficiently. It successfully uncovered 92 bugs, of which 30 have already been confirmed or fixed by developers, highlighting its practical impact on improving the robustness of web-based DL frameworks.

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

Received 20 February 2007; revised 12 March 2009; accepted 5 June 2009

