# OpenReview forum: "TensorJSFuzz: Effective Testing of Web-Based Deep Learning Frameworks via Input-Constraint Extraction"
_ACM.org/TheWebConf/2025/Conference — WWW 2025 Poster_

### Official Review · Reviewer_jLtU · 2024-11-08

**Novelty:** 3
**Technical Quality:** 4

**Review:**

The paper titled "TensorJSFuzz" aims to address a critical gap in the testing of web-based deep learning frameworks, particularly focusing on the need for systematic evaluation methods for platforms like TensorFlow.js. While the topic is relevant and timely, the execution leaves much to be desired.

- Pros
1. The integration of deep learning into web applications is timely, and the paper identifies a significant need for effective testing methodologies.
2. Introducing the concept of input-constraint extraction is a positive step, as it lays groundwork for further exploration in this domain.

- Cons
1. The technical details provided in the paper are insufficient to warrant confidence in the proposed methodology. Key aspects of the input-constraint extraction process are not thoroughly explained, leaving readers unclear about the implementation and potential limitations. The reliance on GPT for results raises concerns about the originality and complexity of the approach.
2. While the paper introduces an approach to testing, it does not significantly advance the field. The methods discussed seem to borrow heavily from existing frameworks without offering new insights or techniques that could enhance the reliability of web-based DL systems.
3. The paper fails to convincingly argue the importance of its research. Although the necessity of testing web frameworks is acknowledged, the authors do not provide compelling evidence or case studies demonstrating the potential consequences of inadequate testing in this domain. This lack of urgency diminishes the impact of their findings.
4. The experiments conducted to validate the proposed methodology lack rigor. The outcomes derived from GPT-based methods do not provide enough evidence of effectiveness in real-world scenarios.
5. The paper fails to discuss how the proposed framework could be integrated into existing development processes, limiting its practical relevance and applicability for developers.

**Questions:**

1. Could you provide concrete examples or case studies that illustrate the potential risks associated with inadequate testing of web-based DL frameworks?
2. What criteria did you use to evaluate the effectiveness of your methodology? Are there plans for more extensive testing in real-world applications?
3. How do you envision your proposed framework being integrated into current development workflows? What steps would developers need to take to adopt it?

**Reviewer Confidence:**

3: The reviewer is confident but not certain that the evaluation is correct

**Scope:**

3: The work is somewhat relevant to the Web and to the track, and is of narrow interest to a sub-community

---

### Official Review · Reviewer_Z7M7 · 2024-11-23

**Novelty:** 5
**Technical Quality:** 5

**Review:**

The paper presents a fuzzing tool tailored for testing web-based deep learning (DL) frameworks, particularly TensorFlow.js. The approach leverages LLMs to extract input constraints from source code, enabling the generation of high-quality inputs that bypass validation checks and trigger potential backend inconsistencies. The experimental evaluation demonstrates the tool's superiority in generating valid inputs and detecting bugs compared to existing methods. It is good to see the use of LLMs in various fields. The methodology is systematically explained. At the same time, here are some areas for improvement.

It would be better to add some details on how the proposed approach is generalizable instead of simply a footnote statement. It is fine to put them in the Appendix if there are concerns regarding the space. This will strengthen the contributions of the proposal.

The constraint extraction currently relies on the out-of-box GPT-4. I would suggest testing with other LLM models. Considering fine-tuning a model for this fuzzing framework would be even better.

**Questions:**

1. Could you provide an example of LLM outputs (before any post-processing)?

2. LLM outputs can be random. For each run, did you run the LLM component once or multiple times? Did you consider some ways to examine the quality of LLM outputs?

3. How robust is the approach to updates or changes in the capabilities of the LLMs used (e.g., GPT-4)?

4. What are the specific challenges in adapting TensorJSFuzz to other web-based DL frameworks (e.g., ONNX.js or ml5.js)?

5. What is the scalability of the approach when testing DL models with a higher number of operators or extremely large tensor inputs?

6. Were there any cases where detected "bugs" were deemed non-issues or platform-specific quirks? How were these filtered out?

**Reviewer Confidence:**

1: The reviewer's evaluation is an educated guess

**Scope:**

4: The work is relevant to the Web and to the track, and is of broad interest to the community

---

### Official Review · Reviewer_5R7x · 2024-11-25

**Novelty:** 5
**Technical Quality:** 5

**Review:**

Pro.
- This paper presents a clear structure, a fuzzy testing framework based on constraint extraction and test data generation, and achieves more efficient bug finding by generating high-quality input use cases.

- Experimental results in the paper show that TensorJSFuzz outperforms existing baseline methods in generating valid inputs and identifying bugs. This proves the potential and effectiveness of this approach in practical applications, especially in improving the test coverage and bug detection capabilities of Web-based DL frameworks.

- The authors not only propose the method, but also conduct an exhaustive evaluation, including comparison with prior art, analysis of bug types, and manual verification of the quality of extraction constraints. This comprehensiveness provides the reader with an opportunity to gain insight into the performance of the approach.

Cons.
- The Introduction phase should clarify the differences between the server-side DL framework and the web-side Dl framework to better clarify the differences between the contributions of this paper and previous work.

- Not enough ablation experiments for constraint extraction and input generation to quantitatively evaluate the contribution of each method.

**Questions:**

- The author should list the test framework used to test Tensorflow/Pytorch deployed on the server side, the same code test object, or explain why it cannot be applied to Tensorflow.js. The author mainly emphasizes the efficiency of program execution caused by the difference in execution environment, but it does not explain the difference in code style well. For example, the second paragraph of Intro can supplement specific examples.
- Contrast the fuzzy test framework. For example, in the experiments in Table 2 of Sec.4.2, DocTer is a constraint rule based on document API. Why is the dependency extraction introduced by LLM not considered on the basis of DocTer, so as to determine which method 3.1.1 and 3.1.2 plays a greater role in constraint extraction?
- Generate ablation experiments for test cases. Two methods of input generation and adjustment are mentioned in 3.2. The author should make clear the influence of these two methods on the correctness and efficiency of finding bugs in the group in the Sec 4.3 experiment. At the same time, the influence of the input generation method mentioned in the DocTer + article on the final fuzzy test effect should also be evaluated. Since DocTer is an input generation for document learning based on the mainstream DL framework, this comparison is not fair to the Baseline DocTer.
- Figure 6. First appearance of abbreviation BNF: Backus-Naur Form

**Reviewer Confidence:**

3: The reviewer is confident but not certain that the evaluation is correct

**Scope:**

4: The work is relevant to the Web and to the track, and is of broad interest to the community

---

### Official Review · Reviewer_SxXm · 2024-11-26

**Novelty:** 5
**Technical Quality:** 4

**Review:**

To address the issue of low-quality test case generation for existing web-based deep learning (DL) frameworks, which fails to accurately extract constraint information from the source code of DL operators, the authors propose the TensorJSFuzz method. This approach utilizes large language models (LLMs) to extract input constraints from the source code of DL operators. It employs a generation technique that includes type-aware input generation and dependency-aware input refinement, effectively generating diverse and high-quality test inputs. This work presents a new perspective on testing deep learning frameworks.
Experimental results demonstrate that TensorJSFuzz outperforms baseline methods in generating valid inputs and identifying errors. Specifically, out of the 92 errors detected, 30 have been confirmed or fixed by developers, highlighting its effectiveness in improving the robustness of deep learning frameworks.

The structure of this paper is clear, and the logic is relatively rigorous. The originality of this article is reflected in the following aspects:
**Methodological Innovation:** TensorJSFuzz is the first testing framework specifically designed for TensorFlow.js, filling a gap in existing testing methods for web deep learning frameworks.

**Technological Innovation:** It utilizes LLMs to extract input constraints and introduces type-aware and dependency-aware generation techniques, enhancing the quality of generated test inputs.

### Pros
1. This paper is the first to apply LLMs to assist in the testing of web-based deep learning frameworks. It leverages the advantages of LLMs in processing and understanding natural language. With the assistance of LLMs, TensorJSFuzz can better extract constraint information from the source code, and the experimental results corroborate this finding.

2. The paper generates high-quality seeds and demonstrates the effectiveness of the method through extensive experiments. The number of detected errors is significant, and the detection efficiency is also very high.

### Cons
1. The constraint extraction section of this paper heavily relies on the performance of the LLM. The method of extracting constraints from source code using the LLM's reasoning capabilities may be affected by the model's accuracy and performance. If the LLM cannot perfectly understand the source code, it may lead to insufficient extraction of input constraints.

2. Although this paper detects many bugs, the proportion of duplicate bugs is relatively high, and the number of newly discovered bugs is low. The tool's ability to detect new bugs in the future remains to be observed.

3. The experimental section of this paper lacks a discussion on the two generation techniques. For example, I would like to know how type awareness and dependency awareness each contribute to and enhance the detection process.

4. The experimental section of this paper does not show the detection effectiveness of different testing oracles. From the bug analysis, it seems that bugs related to crashes and memory errors exhibit a correlation with dependencies, while bugs found through differential testing do not appear to be related to dependencies.

**Questions:**

1.The experimental section could separate section 4.3.2 as an independent research question (RQ) to analyze the categories of bugs found by the method and its uniqueness compared to other methods.

2. The experimental section lacks a comparison between type-aware and dependency-aware generation techniques. It should address how the performance differs between methods that only utilize type awareness or only dependency awareness compared to methods that adopt both techniques, and whether the errors detected are closely related to these two awareness techniques.

3. The experimental section is missing an ablation study on different testing oracles. It should investigate whether different oracles are all effective.

4. Will there be consideration for training and fine-tuning the LLM in the future? For example, using source code and corresponding constraint relationships as training data to further improve the accuracy of constraint extraction.

5. Regarding Figure 6, I do not fully understand the relationship between the syntax of Figure 6 and the constraints extracted by the LLM. Could the authors please explain in detail how to manually check the extracted constraints and summarize the process into a simplified constraint syntax?

6. In the first paragraph of section 3.2, the paper states that traditional methods face high costs when dealing with tensor-related constraints, while the lightweight generation method proposed in this paper does not have this issue. In the experimental section, could the authors clearly describe what high costs mean and compare the costs of different methods through experiments?

**Reviewer Confidence:**

3: The reviewer is confident but not certain that the evaluation is correct

**Scope:**

4: The work is relevant to the Web and to the track, and is of broad interest to the community

---

### Official Review · Reviewer_7R1A · 2024-12-02

**Novelty:** 5
**Technical Quality:** 5

**Review:**

This paper presents TensorJSFuzz, a novel method for testing web-based DL frameworks to extract constraints directly from the source code of framework APIs to ensure input quality. Also, TensorJSFuzz performs type-aware random generation coupled with dependency-aware refinement to create high-quality test inputs by leveraging Large Language Models to understand the code and extract input constraints. The experimental results validates its effectiveness in improving the robustness of web-based DL frameworks. This paper is well-written, and easy to follow.

**Questions:**

This work only chooses ChatGPT for constraint extraction using a one-shot prompting strategy. I am curios that the proposed method apply to the other LLM?

**Reviewer Confidence:**

1: The reviewer's evaluation is an educated guess

**Scope:**

4: The work is relevant to the Web and to the track, and is of broad interest to the community